# Investigation on Subjects' Seasonal Perception and Adaptive Actions in Naturally Ventilated Hostel Dormitories in the Composite Climate Zone of India

**Sanjay Kumar [1], Manoj Kumar Singh [2,3,]\*, Nedhal Al-Tamimi [4], Badr S. Alotaibi [4] and Mohammed Awad Abuhussain [4]**

1   Mechanical Engineering Department, Dr. B R Ambedkar National Institute of Technology, Jalandhar 144011, Punjab, India; sanjay@nitj.ac.in
2   National Institute of Technology Arunachal Pradesh, Jote, Papum Pare 791123, Arunachal Pradesh, India
3   Faculty of Civil and Geodetic Engineering, University of Ljubljana, Jamova Cesta 2, 1000 Ljubljana, Slovenia
4   Architectural Engineering Department, College of Engineering, Najran University, Najran 66462, Saudi Arabia; naaltamimi@nu.edu.sa (N.A.-T.); bsalotaibi@nu.edu.sa (B.S.A.); maabuhussain@nu.edu.sa (M.A.A.)
\*   Correspondence: mksinghtu@gmail.com

**Abstract:** A seasonal adaptive thermal comfort study was done on university students in naturally ventilated dormitories in the composite climate zone of India. A total of 1462 responses were collected from the students during the field study spread over the autumn, winter, spring, and summer seasons of the academic year for 2018 and 2019. A "Right Here Right Now" type of surveying method was adopted, and the indoor thermal parameters were recorded simultaneously using high-grade instruments. The subjects' mean thermal sensation (TS) was skewed towards a slightly cool feeling for the combined data. Most occupants preferred a cooler thermal environment during the summer season, while hostel residents desired a warmer temperature during autumn, winter, and spring seasons. During the summer season, the PMV−PPD model overestimated the subjects' actual thermal sensation, while it underestimated the their thermal sensation in the winter season. The mean comfort temperature $T_{comf}$ was observed to be close to 27.1 (±4.65 °C) for the pooled data. Mean clo values of about 0.57 (±0.25), 0.98 (±0.12), 0.45 (±0.27), and 0.36 (±0.11) were recorded during the autumn, winter, spring, and summer seasons, respectively. Furthermore, switching on ceiling fans and opening doors and windows improved occupants' thermal satisfaction during different seasons. The study results show the effective use of environmental controls and the role of thermal adaptation in enhancing the subjects/overall thermal satisfaction in the composite climate of India.

**Keywords:** field surveys; thermal perceptions; adaptive actions; hostel dormitories; composite climate of India

## 1. Introduction

In the last two decades, an increasing trend in the building sector's energy consumption has been observed worldwide, contributing to more than 40% of the total global greenhouse emissions. With the ever-increasing expectation of the indoor thermal environment, this is projected to increase further in the near future [1]. In India, the building sector is the second-largest contributor to greenhouse emissions and overall energy consumption [2]. In the context of India, many educational buildings exist and are emerging as the leading sector in overall building energy consumption [3]. A large chunk of the energy is consumed by buildings to restore thermal comfort. According to ASHRAE Standard 55 and ISO 7730, thermal comfort is the condition of the subject's mind that expresses thermal satisfaction with the thermal environment surrounding them. A questionnaire-based subjective evaluation methodology is generally adopted to evaluate the thermal comfort conditions in different building types [4,5]. International standards such as ASHRAE Standard 55 [4] and

ISO 7730 [5] are widely used to assess thermal comfort conditions in built environments. Primarily, ASHRAE Standard 55 [4] and ISO 7730 [5] are based on Fanger's heat balance model (PMV/PPD) [6]. Over the last two and half decades, numerous field studies carried out by researchers have shown that the PMV/PPD approach fails to capture the entire spectrum of parameters associated with the psychological, physiological, and socio-cultural aspects. These parameters play a vital role in the adaptation mechanism the occupants of different built environments go through, thus impacting their thermal comfort perception and expectations [7–12]. The above-mentioned parameters of thermal comfort become part of the study when the adaptive thermal comfort principle is applied to record the thermal response of the subjects in the field surveys. Adaptive thermal comfort through various field studies has shown massive potential in evaluating the indoor thermal environment and minimizing buildings' energy consumption without compromising occupant comfort [13].

Recently, researchers have reported several comprehensive reviews [14–17] helping to understand the causes of individual differences and thermal adaptation mechanisms of human subjects in different buildings, climates, ventilation strategies, and other contextual factors of thermal comfort. With the availability of the most extensive thermal comfort data in the ASHRAE Global Thermal comfort database, researchers have pointed out that thermal adaptation may significantly define the thermal acceptability ranges in different buildings and climates [18–20]. In India, research on thermal comfort gained popularity after the pioneering work of Nicol [21], and Sharma and Ali [22] in the late 1980s. After this, many researchers started evaluating the thermal adaptation mechanism of Indian inhabitants, considering the country's diverse climatic and geographical diversity. For instance, Indraganti [23] conducted a series of field monitoring in multi-story residential apartments in Southern India and stated that regional static comfort limits were not applicable to define the thermal comfort needs of residents in the hot and humid climate of Southern India. The author also analyzed adaptive strategies to achieve thermal comfort, such as windows, balconies, the use of external doors, fans, and clothing adaptations [24]. Singh et al. [9,25] also reported results related to the thermal adaptation of residents in vernacular houses in the north-east part of India. The authors argued that an adaptive approach to thermal comfort is more suitable for analyzing the thermal adaptation of people under different climatic zones of this region. The National Building Code of India [26] has adopted the India Model for Adaptive Comfort (IMAC) [27] to define the 80% and 90% thermal acceptability ranges of thermal comfort for naturally ventilated and mixed-mode buildings in different climatic zones of India. Since then, adopting similar approaches and methods, researchers in India have conducted field studies considering different building types, i.e., classrooms [28,29], university buildings [30], offices [27,31,32], residential buildings [23,25], hostel dormitories [33–35], and special metabolic activity spaces [36]. Researchers have concluded that thermal comfort is a complex phenomenon and depends on the different adaptation mechanisms and contextual factors inculcated in the adaptive approach to thermal comfort.

Educational buildings and the associated built environment play a significant role in students' learning and wellbeing [3,17]. Students in university generally fall in the age group of 18–26 years old and spend a lot of time in hostel dormitories for their undergraduate or postgraduate studies. Thus, emphasis should be placed on designing and constructing hostel dormitories so that they provide a conducive and quality thermal environment to stimulate the learning process [29,30], without compromising students' needs of comfort and health [37]. Moreover, the indoor thermal environment and air quality in hostels are very different from other building types because of significant differences in the age groups, occupancy patterns, behaviours, and activities carried out by students [34,37]. Considering the importance of a quality built environment in students' learning process, many researchers have carried out field studies to investigate the thermal performance of educational buildings in different climates and their relation to occupants' overall comfort requirements. For instance, Dalhan et al. [37] conducted a research study on thermal comfort in three high-rise hostel buildings in Malaysia's hot and humid climate.

They found that the mean neutral temperature of hostel buildings was close to 28.8 °C. Lai [38] used gap theory for a post-occupancy evaluation in order to explore the role of six parameters, namely visual comfort, acoustic comfort, fire safety, hygiene, and information and communication technology, in students' expectation and satisfaction. In India, Dhaka et al. [33] carried out a field study in a hostel building in a composite climate during the peak summer season in Jaipur City. The study found a higher comfort bandwidth of approximately 7.9 °C and a mean comfort temperature of 30.2 °C. Kumar et al. [34,35] carried out a questionnaire-based field study in naturally ventilated hostel buildings in Jaipur and Jalandhar City during the autumn and winter seasons, and compared the results. The study found that students living in the hostel had different comfort expectations that those in office or residential buildings. Mean Griffiths comfort temperatures ($T_c$) of 30.4 °C and 29.7 °C were observed for Jaipur and Jalandhar City, respectively. The data analysis also showed the extension of comfort boundaries by 1.8 °C at a high airspeed (ceiling fans).

University buildings consist of different built environments such as offices, residential buildings, classrooms, lecture theatres, and hostel dormitories. In India, researchers have carried out thermal comfort studies and found the expectation and preferences of occupants in offices, residential buildings, and classrooms during the summer and winter seasons for different climatic zones [9,22,27]. However, a literature review carried out here by the authors showed that very few studies have been done in hostel dormitories that have highlighted the subject's behavioral adaptation, use of controls, and thermal adaptation in different seasons of the composite climate of India. Therefore, the present study systematically investigates the thermal preferences and sensations of residents during different seasons in hostel dormitories under the composite climate of India. Furthermore, the study reports the behavioral adaptations and environmental controls of the occupants for their thermal comfort requirements.

## 2. Methodology

### 2.1. Location and the Selected Hostel Buildings

A questionnaire-based field study was done in naturally ventilated dormitory buildings at the National Institute of Technology premises, Jalandhar (latitude = 31.3° N, longitude = 75.58° E, mean sea level = +228 m). Jalandhar City is in the state of Punjab and is in the composite climate of India. The composite climate zone has a large geographical spread, so it has more climatic diversity than other climatic zones of India. It has four distinct seasons, i.e., winter (November–February), spring (March), summer (April–September), and autumn (October) [31]. The summer season is spread over six months. It is characterized by scorching and dry weather conditions and a maximum temperature exceeding 45 °C, while, during the winter season, the outdoor temperature dips below 2 °C. Figure 1 shows the recorded outdoor temperature and relative humidity profile in different months of the year at the study location. It can be seen clearly that during the summer season, the air temperature peaks start from March (mean temperature = 29 °C) and attain a maximum temperature during May (mean temperature = 35 °C) and June (mean temperature = 30.3 °C) at the study location. The relative humidity is generally very low during the summer season, and the months are mostly dry. Following June, July and August are considered rainy months, characterized by a high relative humidity and low air temperature. Autumn and spring have generally moderate ambient conditions with a mean air temperature not exceeding 25 °C. The winter season consists of December and January months under the composite climate of India. During the winter season, the minimum air temperature falls below 2 °C, with an average temperature range between 15–22 °C with moderate relative humidity conditions.

A naturally ventilated dormitory environment with students in typical clothing on a typical survey day is shown in Figure 2. The investigated hostel dormitories were multi-story buildings and were constructed using high thermal capacity construction materials, i.e., concrete mixture, brick burned, plaster, etc. The roofs of the dormitories were made up of reinforced concrete cement (RCC) with a thickness of ~0.15 m, with ~0.015 m thick

gypsum plaster on both sides. The external walls of the hostel buildings had a thickness of about 0.20−0.23 m. The window assemblies consisted of a single piece of clear glass of ~0.003 m thickness with a U-value of ~5.7 W/m²K.

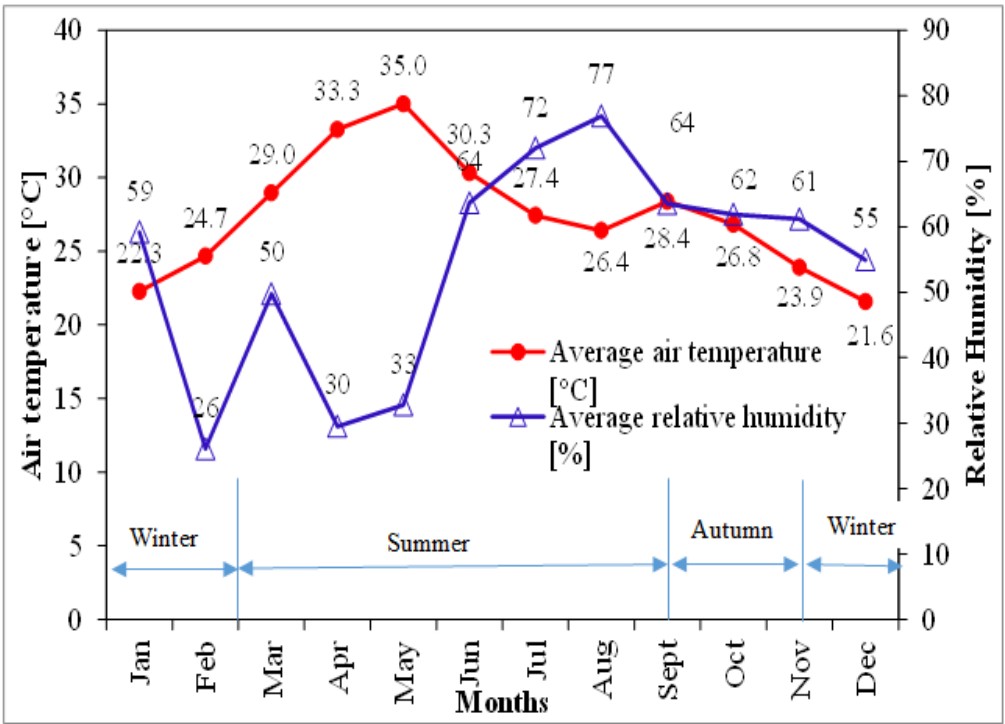

**Figure 1.** Ambient climatic parameters for different months at the location.

*2.2. Sample Size Description*

A total of 1462 questionnaire-based responses were returned during the field study. The subjects were undergraduate and postgraduate students with a mean age of about 20 years and were healthy individuals. The field study was spread over different seasons of the academic year for 2018 and 2019. Therefore, the number of subjects who participated in the survey varied in each season, as shown in Table 1. Furthermore, all of the subjects voluntarily participated in the field study.

**Table 1.** Information of the selected location, sample size, and gender (N = 1462).

| City | Location | Season | No of Samples |
|---|---|---|---|
| Jalandhar, India | Latitude—31.33° N, Longitude—75.58° E, Altitude—228 m | Autumn Winter Spring Summer | 135 181 248 898 |

*2.3. Field Study and Survey Protocols*

The "Right Here Right Now"-based questionnaire was employed to record the students' thermal sensation votes and preferences. The questionnaire used in the field study is provided as "Appendix A". The ASHRAE seven-point thermal sensation scale and Nicol's [39] five-point thermal preference scale were used to record the subjects' thermal sensations and thermal preferences in the indoor environment (Table 2). Laboratory-grade industry calibrated instruments, with a high precision and accuracy, used in the field study are shown in Figure 2d. The make, range, and accuracy of the instruments used in the field study are presented in Table 3. During the interaction with the subjects, the indoor thermal parameters were recorded using the instruments placed close to the students and at a height of 1.1 m [4]. The clothing values of each student were estimated using the

insulation values of each clothing ensemble provided in ISO-7730 standard and ASHRAE standard 55-2020 [4,5]. The subject's metabolic rates were calculated according to the checklist provided in the ISO-7730 standard and ASHRAE standard 55-2020.

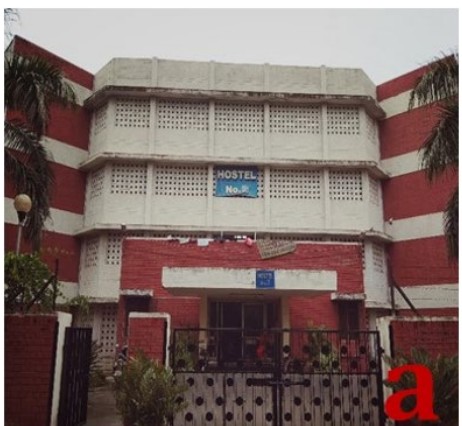
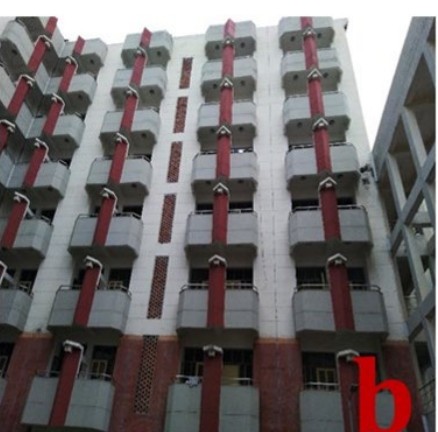
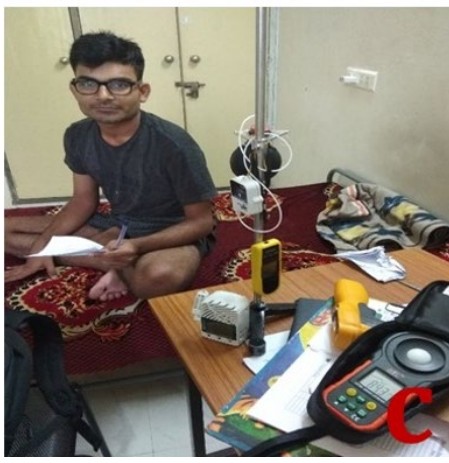
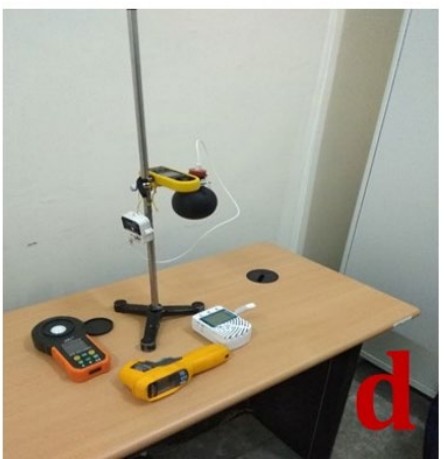

**Figure 2.** (**a**,**b**) Pictures of studied dormitories, (**c**) typical survey environment, and (**d**) instruments used to record the thermal parameters during the field study.

**Table 2.** Sensation and preference scales used in the present study.

| Scale Values | Thermal Sensation | Thermal Preference | Overall Comfort |
|:---:|:---:|:---:|:---:|
| +3 | Hot | | |
| +2 | Warm | | |
| +1 | Slightly warm | Cooler | Uncomfortable |
| 0 | Neutral | No change | Comfortable |
| −1 | Slightly cool | Warmer | |
| −2 | Cool | | |
| −3 | Cold | | |

**Table 3.** Make, range, and accuracy of instruments used in the field study.

| Description | Make of Instruments | Parameter Used | Range | Accuracy |
|:---:|:---:|:---:|:---:|:---:|
| Thermo-hygro $CO_2$ meter | TR—76Ui | Air temperature<br>Relative humidity<br>$CO_2$ level | 0–55 °C<br>10–95% RH<br>0–9999 ppm | ±0.5 °C<br>±5% RH<br>±50 ppm ± 5% |
| Globe thermometer | Tr-52i, globe (dia. 75 mm) | Globe temperature | −60–155 °C | ±0.3 °C |
| Infrared thermometer | Fluke 61 | Surface temperature | −18–275 °C | ±2 °C |
| Thermal anemometer | Testo-405 | Air velocity<br>Air temperature | 0.01–10.00 m/s<br>−20–50 °C | 0.01 m/s<br>±0.1 °C |

## 3. Results and Discussion

### 3.1. Indoor and Outdoor Thermal Environmental Conditions

Table 4 presents the descriptive statistical summary of measured indoor thermal environment parameters during the field study. The mean indoor air temperature varied between 17.3 °C to 30.8 °C, and the mean indoor relative humidity varied between 30–78% from winter to summer at the study location. The average air was recorded to. Be higher during the autumn and summer seasons than in spring and winter. The mean airspeed was about 0.71 m/s for the combined dataset, and this value is well within the limits of no paper blowing conditions as defined in ASHRAE Standard 55-2020 [4].

**Table 4.** Statistics of the indoor thermal parameters measured in different seasons.

| Parameters | Autumn | | Winter | | Spring | | Summer | | All Season Data | |
|---|---|---|---|---|---|---|---|---|---|---|
| | Mean | sd | Mean | sd | Mean | sd | Mean | sd | Mean | sd |
| $T_a$ | 25.6 | 2.52 | 17.3 | 1.8 | 21.5 | 3.5 | 30.8 | 2.9 | 27.3 | 5.6 |
| $T_g$ | 24.6 | 2.69 | 16.6 | 1.6 | 20.8 | 3.5 | 30.3 | 2.5 | 26.7 | 5.7 |
| $T_{out}$ | 28.7 | 3.45 | 14.5 | 2.8 | 24.5 | 2.8 | 35.4 | 3.7 | 29.4 | 6.4 |
| $Rh_i$ | 48.9 | 6.5 | 61.5 | 8.8 | 54.1 | 10.7 | 61.1 | 16.0 | 59.1 | 15.4 |
| AS | 0.29 | 0.47 | 0.13 | 0.28 | 0.18 | 0.30 | 1.06 | 0.72 | 0.71 | 0.69 |

N: no. of samples; $T_a$: indoor air temperature (°C); $T_g$: indoor globe temperature (°C); $RH_i$: indoor relative humidity (%); AS: airspeed (m/s).

### 3.2. Analysis of Seasonal Thermal Sensation Votes and Preference Votes

As the field study was carried out across the different seasons of the year, a significant variation in the measured indoor and outdoor thermal parameters was observed. The thermal sensation voting patterns of the surveyed subjects during different seasons and from the pooled data are shown in Figure 3a,b. From the figure, a proportionally higher number of subjects voted "slightly cool", "cool," neutral", and "slightly warm" during the autumn, winter, spring, and summer seasons, respectively (Table 5). The subject's mean thermal sensation skewed towards "slightly cool" (mean TS = −0.15; sd = ±1.37) in the pooled dataset. From Figure 3, it can be concluded that the students perceived the existing thermal environment as being "slightly cool" rather than "neutral" in the surveyed dormitories. About 71.2% of subjects voted in three central categories on the thermal sensation scale, i.e., ±1, and can be assumed to be comfortable. Furthermore, 32%, 59%, 23%, and 5% of the subjects voted for the cooler side of the TS scale (TS ≤ −1) during the autumn, winter, spring, and summer seasons. Conversely, only 18.1% of subjects voted for the warmer side (TS ≥ +1) during the summer season.

The mean thermal preference was observed to be +0.98, +0.87, +0.01, and −0.72 during the autumn, winter, spring, and summer seasons, respectively. The positive sign indicates that the subjects preferred to be warmer, and the negative sign indicates that the subjects preferred to be cooler. It can be seen that subjects preferred a cooler thermal environment in the summer season, while a warmer thermal environment was desired by hostel students in the autumn, winter, and spring seasons. A total of 39.5%, 19.9%, 25.5%, and 40.5% of subjects voted for "no change", i.e., "neutral" for the existing thermal environment during the autumn, winter, spring, and summer season at the location. However, 39.2%, 65%, and 31.2% of students preferred warm thermal environments during the autumn, winter, and spring seasons, respectively, whereas only 13% of occupants preferred a cooler thermal environment during the entire study period.

We also recorded the overall thermal comfort of the students in the prevailing thermal environment on a binary scale. The subjects voted on a binary scale, i.e., 1 indicating un-comfortable and 0 indicating comfortable, corresponding to the prevailing indoor thermal environment. Figure 4 shows the subjects' voting patterns regarding overall comfort in different seasons and in the pooled dataset. About 88.8%, 76.2%, 92.3%, and 75.5% of subjects voted "comfortable" in autumn, winter, spring, and summer, respectively. In the pooled

dataset, about 87% of subjects indicated their immediate thermal environment as being comfortable, whereas about 13% of students found their thermal environment uncomfortable.

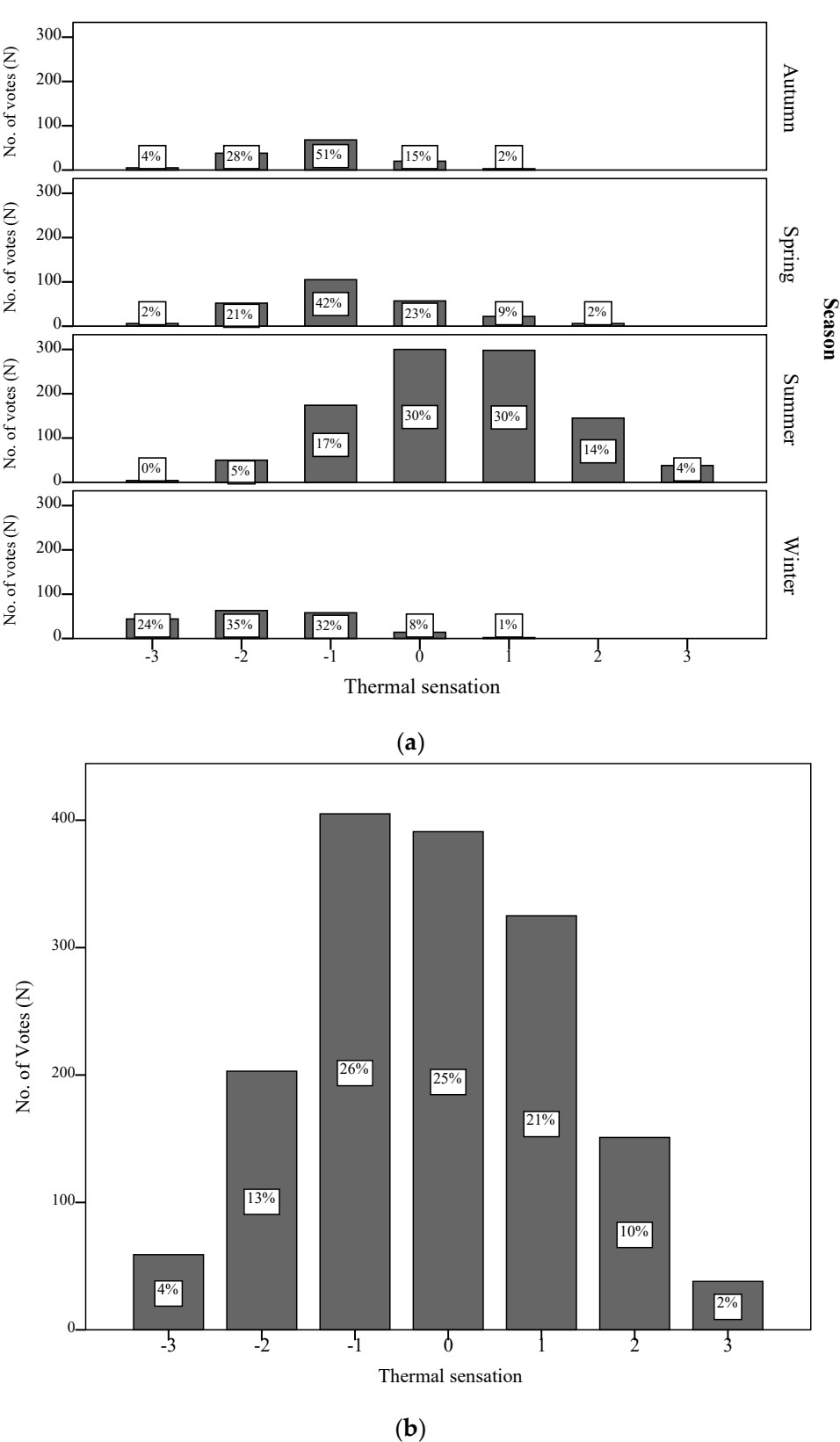

(**a**)

(**b**)

**Figure 3.** Thermal sensation votes distribution (**a**) for different seasons and (**b**) on combined database.

**Table 5.** Statistical summary of subjective and objective comfort parameters in different seasons.

| Season | Thermal Sensation (TS) | | Thermal Preference (TP) | | Overall Comfort (°C) | |
|---|---|---|---|---|---|---|
| | Mean | sd | Mean | sd | Mean | sd |
| Autumn | −1.16 | 0.81 | 0.98 | 0.78 | 0.11 | 0.32 |
| Winter | −1.73 | 0.95 | 0.87 | 0.72 | 0.24 | 0.43 |
| Spring | −0.78 | 1.03 | 0.01 | 0.87 | 0.08 | 0.17 |
| Summer | 0.51 | 1.20 | −0.72 | 0.75 | 0.24 | 0.41 |
| All seasons combined | −0.15 | 1.37 | −0.38 | 0.91 | 0.20 | 0.40 |

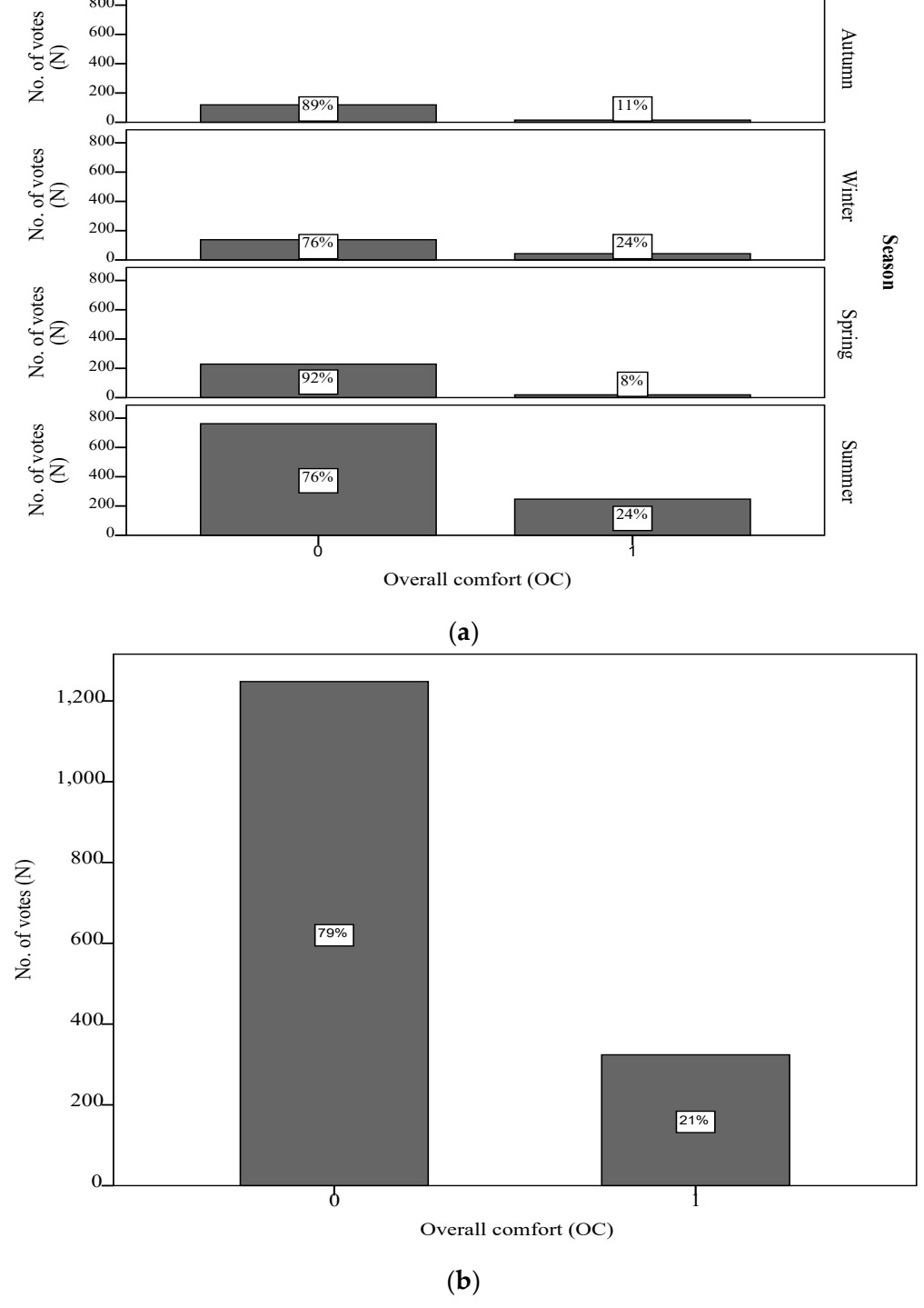

**Figure 4.** Overall comfort votes recorded in (**a**) different seasons and (**b**) in the pooled dataset.

## 4. Characteristics of Seasonal Comfort

### *4.1. Linear Regression and PMV−PPD Model Analysis*

The adaptive thermal comfort principle assumes that people in built environments are not only the recipients, but actively participate and take actions to adapt themselves to the existing indoor environmental conditions through physiological, psychological, and behavioral adaptation in different seasons and climates across the world [7–10]. Therefore, in the current study, the seasonal comfort temperatures of the surveyed students were estimated using the PMV−PPD model, linear regression, and Griffiths approach. The procedure defined in standard ASHRAE 55-2020 calculated the PMV and PPD values [4]. The standard suggests that 80% of occupants will be comfortable within a PMV bandwidth of ±0.5 [4]. It can be seen in Figure 5 and Table 6 that there is a discrepancy between PMV with TSV for different seasons and pooled dataset. It can be concluded from Figure 5 that the PMV values overestimated the actual thermal sensation of subjects during the summer season. In the winter season, the PMV values underestimated the thermal sensation of the subjects (Figure 5).

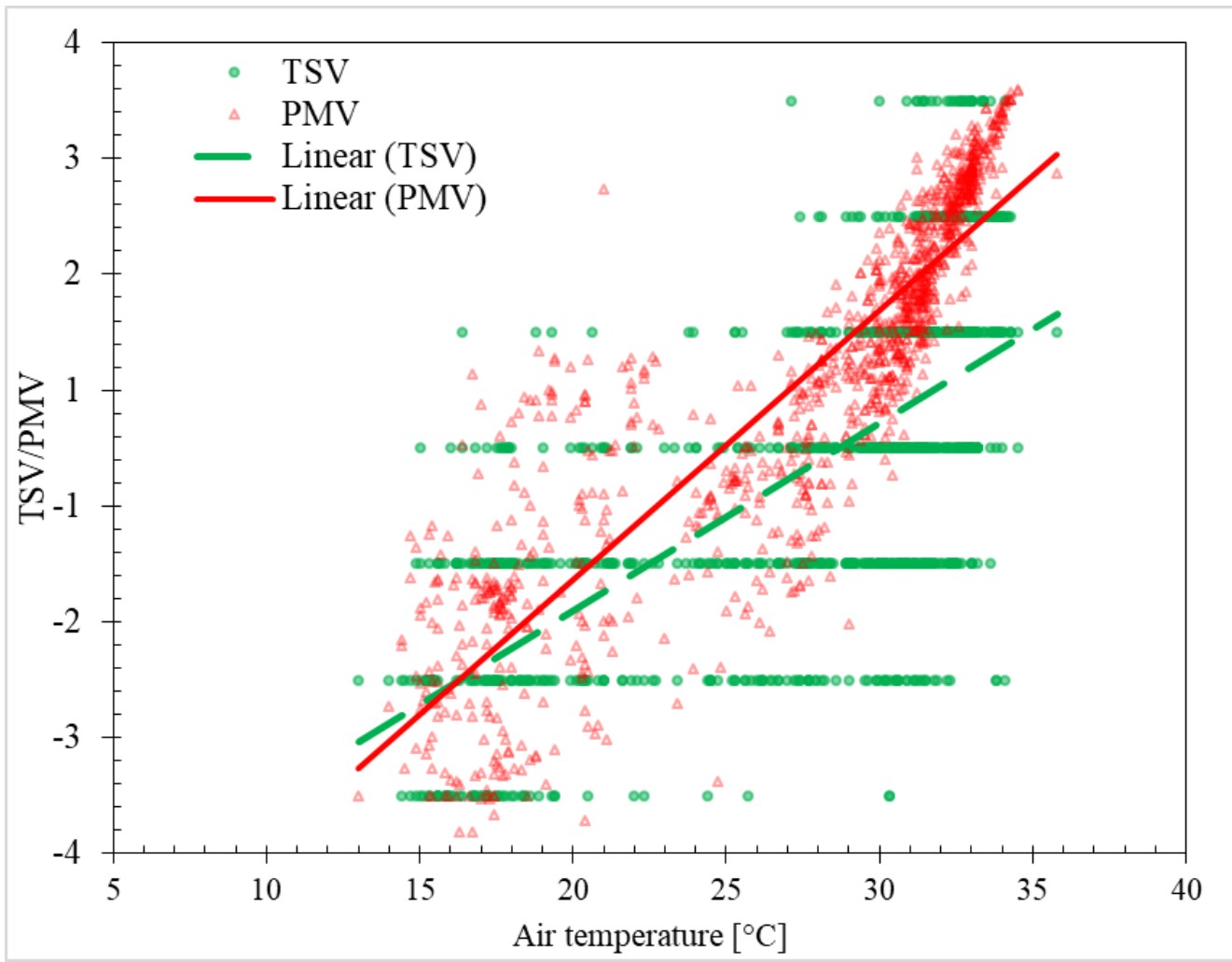

**Figure 5.** Linear regression analyses of PMV votes and TSV.

**Table 6.** Statistics of the linear regression analysis of TSV and PMV on a seasonal basis.

| Case | N | Regression Models * | $R^2$ | $T_n$ (°C) | Mean $T_{comf}$ ± sd (°C) |
|---|---|---|---|---|---|
| All Season data | 1462 | TSV = 0.12 $T_a$ − 3.12 | 0.11 | 26 | 27.1 0 ± 4.6 |
| | | PMV = 0.12 $T_a$ − 3.24 | 0.21 | 27 | |
| Autumn | 135 | TSV = 0.08 $T_a$ − 3.18 | 0.06 | 39.7 | 26.9 ± 2.68 |
| | | PMV = 0.07 $T_a$ − 3.12 | 0.09 | 44.5 | |
| Winter | 181 | TSV = 0.13 $T_a$ − 4.01 | 0.06 | 30.8 | 19.9 ± 2.11 |
| | | PMV = 0.14 $T_a$ − 3.96 | 0.13 | 28.3 | |
| Spring | 248 | TSV = 0.12 $T_a$ − 3.41 | 0.17 | 28.4 | 22.4 ± 3.2 |
| | | PMV = 0.12 $T_a$ − 3.38 | 0.22 | 28.2 | |
| Summer | 898 | TSV = 0.12 $T_a$ − 3.22 | 0.05 | 26.8 | 29.5 ± 2.6 |
| | | PMV = 0.12 $T_a$ − 3.54 | 0.11 | 29.5 | |

N = sample size; TSV = thermal sensation vote; $T_a$ = indoor air temperature; $T_n$ = regression neutral temperature; $T_{comf}$ = Griffiths comfort temperature (°C) with 0.50 as a coefficient. * The regression models are all significant at ($p < 0.001$).

The mean indoor air temperature is considered a neutral temperature at which an average subject will vote neutral "0" on the TSV scale [23,25]. Researchers extensively use the linear regression method to estimate the thermal neutrality of surveyed subjects for different building types and in other climates. We used the linear regression approach to calculate the seasonal neutral temperature in hostel dormitories, as depicted in Figure 6. Behavioral adaptation is evident from the low regression coefficient values ($R^2$) [25]. Analyzing the data, it was found that the mean neutral temperature was 26 °C for the pooled data. A higher comfort bandwidth was also noticed, which varied by more than 16 °C (17.6–33.3 °C) from winter to summer for the hostel residents, showing the wider thermal adaptability corresponding to the climatic variations. Interestingly, the findings of the present study are supported by the studies done by Mishra and Ramgopal [30], Dhaka et al. [33], and Dahlan et al. [37].

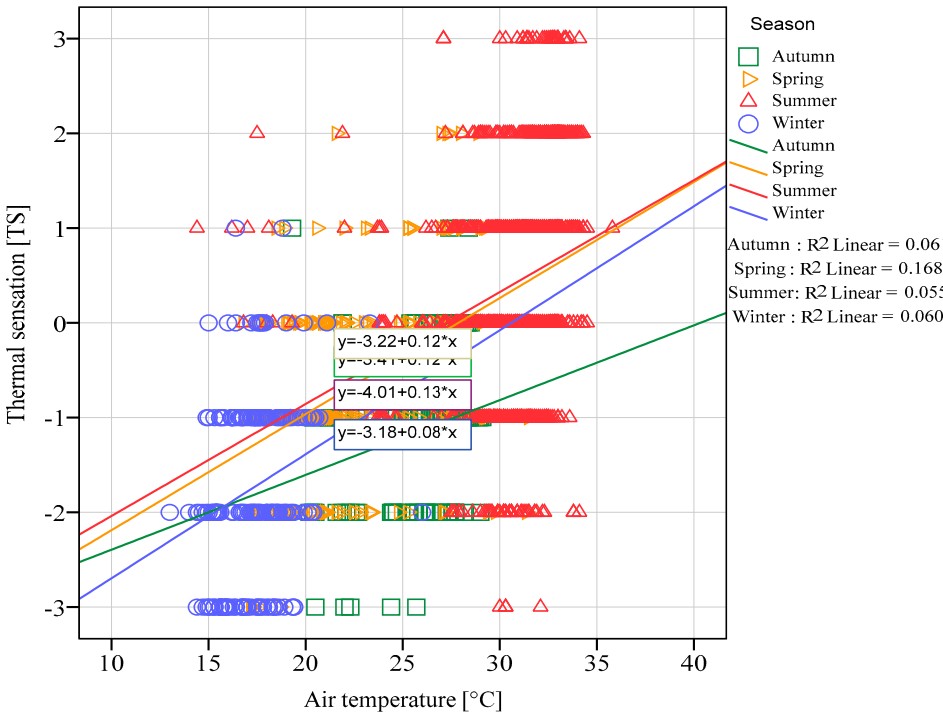

**Figure 6.** Linear regression analysis of TSV for different seasons.

### 4.2. Mean Comfort Temperature ($T_{comf}$): Griffiths Approach

Some researchers have challenged the applicability of the linear regression approach in field studies due to the effects of the adaptive behavior [23,27,40]. Therefore, the mean comfort temperature for each season was estimated using the Griffiths method [41]. The Griffiths equation can be written as follows:

$$T_{comf} = T_a + \frac{[0 - TS]}{GC} \tag{1}$$

where $T_{comf}$ = Griffiths comfort temperature, $T_a$ = air temperature, TS = thermal sensation votes, and GC = Griffiths constant.

Previous studies carried out in the composite climate of India have suggested the use of the Griffiths coefficient of 0.50/°C for the calculation of a neutral temperature [20,27]. Hence, a 2 °C perturbation, i.e., 0.50/°C, was considered for the analysis of the mean comfort temperature ($T_{comf}$) in the present study. The mean comfort temperature ($T_{comf}$) was found to be 27.1 ± 4.65 °C in the pooled data, and is shown in Figure 7. In addition, the mean $T_{comf}$ was about 26.9 ± 2.68 °C, 19.9 ± 2.11 °C, 22.4 ± 3.2 °C, and 29.5 ± 2.6 °C for the autumn, winter, spring, and summer seasons, respectively.

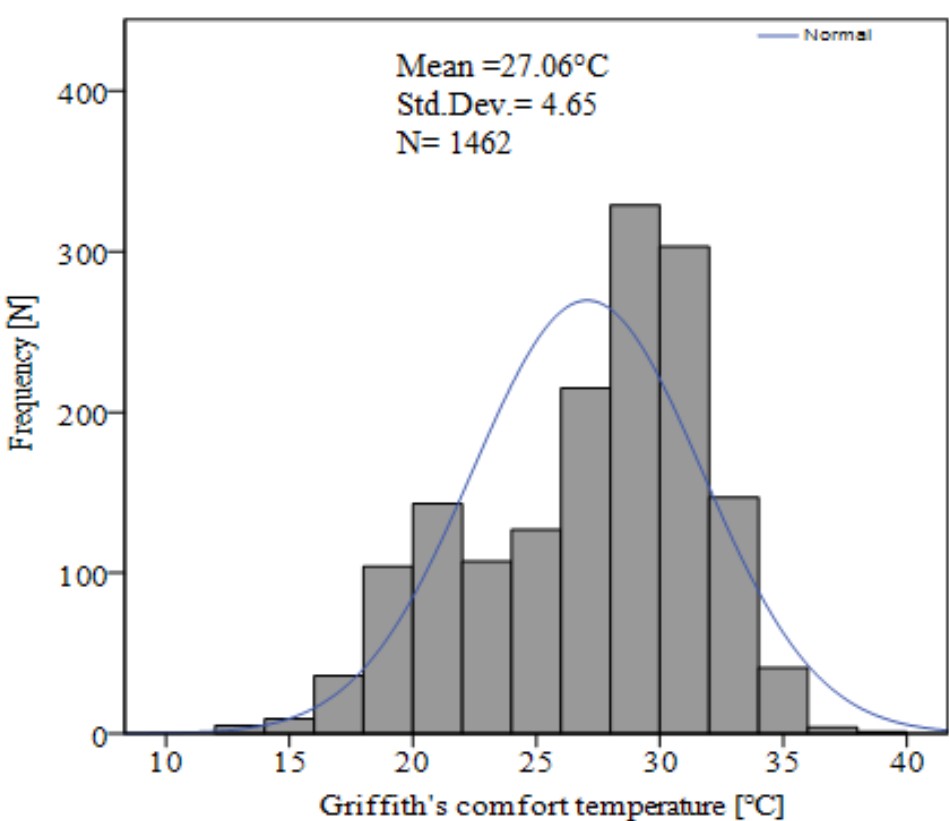

**Figure 7.** Distribution of the calculated mean comfort temperature using the Griffiths method for the pooled data.

## 5. Investigation of Thermal Adaptation Behavior of Residents

The adaptive thermal comfort principle considers that subjects in a built environment are active agents and can exercise various thermal and behavioral adaptations to restore their comfort or make themselves thermally comfortable [7,39]. Therefore, the adaptive behavior of subjects in university hostel dormitories was analyzed in the context of clothing adjustments; the application of environmental controls, i.e., opening and closing of windows and doors, and the use of ceiling fans in different seasons; and in the pooled dataset.

### 5.1. Clothing Adjustments

Students mostly wore clothing ensembles such as shirts/t-shirts and trousers/jeans during the daytime, and nightwear, i.e., pajamas and half sleeve t-shirts, during the holidays and late evening hours. The clothing ensembles ranged between 0.32–1.84 clo for the hot summer season to the cold winter months at the study location. Mean clo values of about 0.57 (±0.25) clo, 0.98 (±0.12) clo, 0.45 (±0.27) clo, and 0.36 (±0.11) clo were recorded during the autumn, winter, spring, and summer seasons, respectively. An average clothing value of about 0.49 ((±0.31) clo was recorded in the pooled database, matching closely with the ASHRAE Standard 55 recommendation for the summer season.

To analyze the characteristic of the adaptation behavior related to the clothing, linear and quadratic regression analyses for indoor air temperature were carried out to observe the inflection points [7,9,31]. Figure 8 shows the linear and quadratic regression fit for predicting the adaptive behavior of students regarding clothing corresponding to the change in indoor air temperature during different seasons at the study location. The inflection points were observed at 18 °C and 31 °C. A sudden change in clothing value was observed at these points, showing adaptation. It can be seen that the correlation coefficient was reasonably strong, which suggests that subjects adaptively use clothing adjustments to restore their comfort. The authors also found similar observations in studies carried out in different climates and building types [29,30,35].

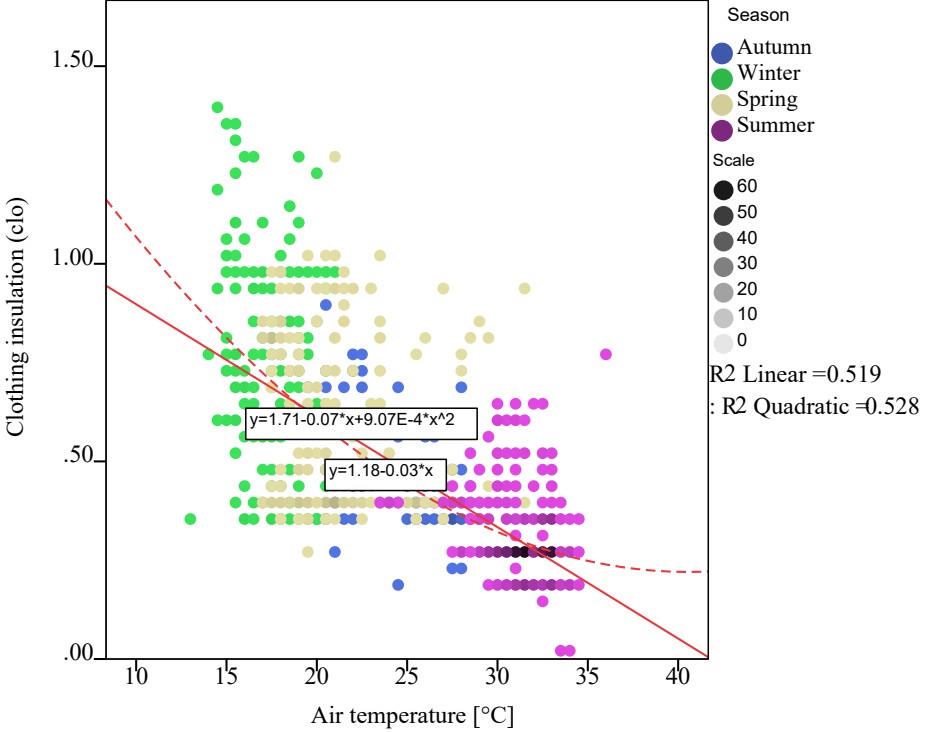

**Figure 8.** Seasonal variations of clothing insulation with indoor air temperature.

### 5.2. Impact of Controls and Exercised Controls on Comfort

Analyzing the use of controls plays an essential role in adaptive thermal comfort studies. The effective use of controls in built environments by subjects enhances thermal comfort and extends the comfort boundaries [16,24,34]. In this context, the authors recorded the use of environmental controls by the students in the dormitory, i.e., windows and ceiling fans, in binary variables (i.e., window open: 1; window closed: 0; fans on: 1; fans off: 0) during the field surveys. Figure 9 shows the percentage of subjects voting feeling comfortable (corresponding to three central categories of the TS scale) when available controls were used. It can be seen that when subjects used the general controls, such as opening windows and doors and switching on ceiling fans, the occupants' thermal comfort

during different seasons improved significantly. In addition, during the summer season, it was observed that more than 80% of students voted feeling comfortable when ceiling fans were operating. In contrast, only 60% of students voted feeling comfortable when windows were open at the time survey.

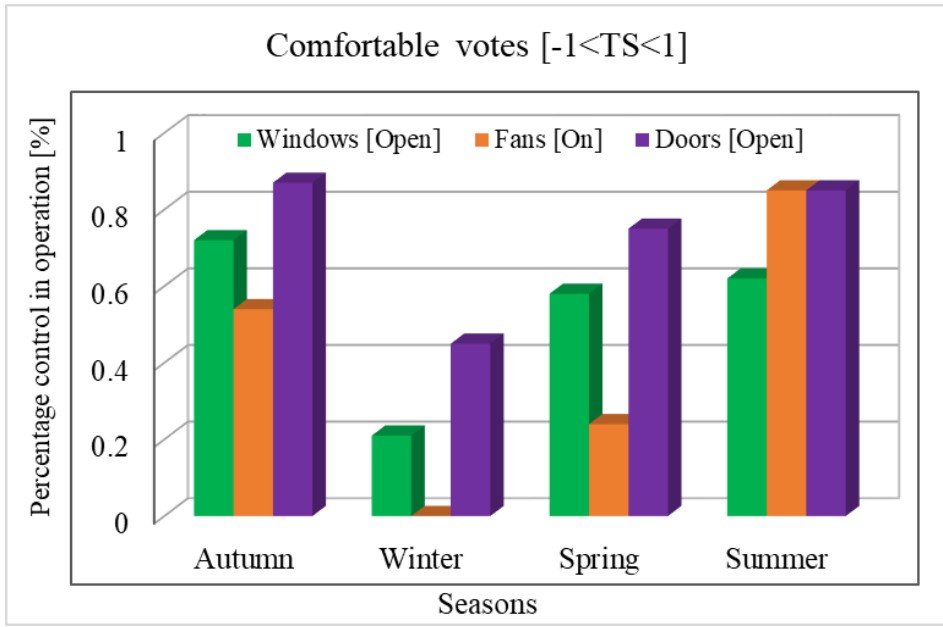

**Figure 9.** Percentage of occupants voting feeling comfortable when environmental controls are in operation during different seasons.

The study results have close resemblances with the finding of other studies conducted under similar climatic conditions but at different locations. Rijal et al. [41] found that about 81% of fans were in use when the indoor air temperature exceeded 28.5 °C in offices at Pakistan. Indraganti et al. [42] noted about 80% fans were in operation at 30 °C in office buildings of India. Manu et al. [43] analyzed the windows and fan use behavior of office occupants based on the field data collected for different climatic zones of India, and concluded that maximum fans as well windows were used under hot and dry, and hot and humid climates of India compared to other climatic zones of India. Similarly, Kumar et al. [31,34] observed that about 50% of windows and 80% of fans were used when indoor air temperature peaked at 28 °C in university buildings situated under the composite climate of India. Singh et al. [44] also predicted a similar observation in office buildings located in the north-east part of India.

To gain more insight, we further plotted the comfortable [±1 TS votes] on the ASHRAE Standard 55-2020 comfort zone when windows and ceiling fans were in operation. ASHRAE Standard 55-2020 [4] graphically defines thermal comfort boundaries on a typical psychrometric chart describing the operative temperature and humidity range for occupants corresponding to the sedentary activity level (1−1.3 met) and clo value in the range of 0.5−1 clo. Furthermore, ASHRAE Standard 55-2020 recommends a maximum indoor airspeed of 0.80 m/s to avoid paper blowing conditions in office buildings. Figure 10a,b shows the plotting of comfortable votes when windows were "open" or ceiling fans were "on" during the field surveys. It was observed that when windows were open, subjects felt comfortable at a high relative humidity and high indoor air temperatures. In addition, the maximum airspeed was recorded at close to 2 m/s during the summer. The subjects voted feeling comfortable even when the indoor air temperature was about 34 °C and the relative humidity was more than 70%. These results are supported by the authors' previous findings under similar climatic conditions for office buildings [8,32,33].



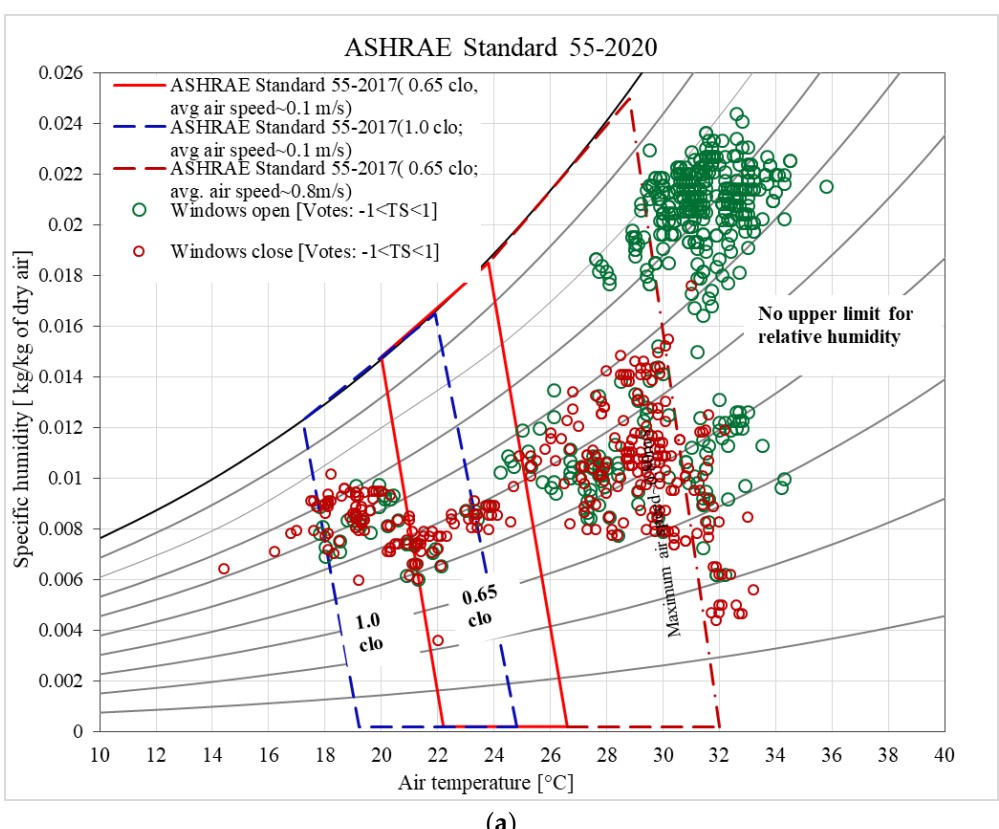

(**a**)

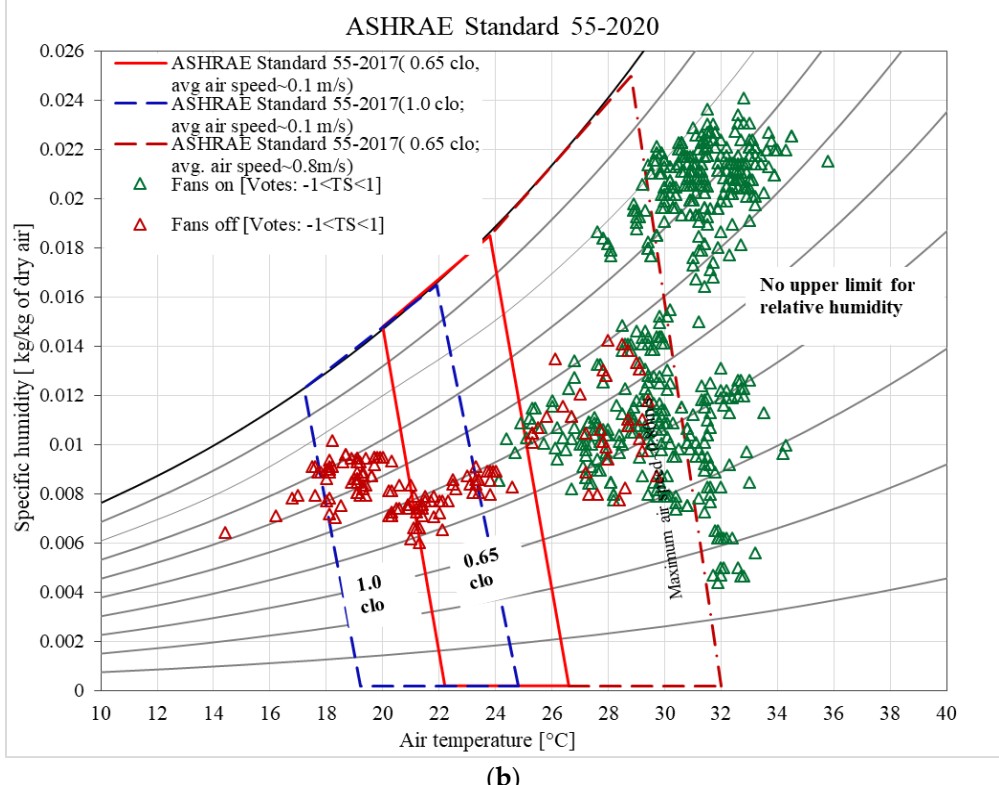

(**b**)

**Figure 10.** Plotting comfortable votes on the ASHRAE Standard 55 – 2020 comfort zone when (**a**) windows are open/closed and (**b**) fans are on/off.

### 6. Summary of Work and Conclusions

In this study, a seasonal comfort study was carried out in naturally ventilated hostel dormitories under the composite climate of India. One of the prime objectives of the built environment is to provide the desired thermal comfort to the occupants. If this is disregarded, occupants make use of mechanical and electrical devices to achieve the desired thermal comfort. This involves costs at various levels and impacts sustainability of the building sector. To improve the sustainability of the building sector, presently, the entire world is working on various issues to reduce the energy consumption. Precise evaluation of the comfort parameters of different built environments and occupants' behaviour characteristics are key for the reduction and optimal use of energy in buildings. To achieve comfort in the built environment, it is of the utmost importance that the comfort parameters of different built environments and occupants' behaviour characteristics must be known by building designers and architectures so that they can design buildings that will provide optimum comfort to the occupants and consume less energy so as to provide the necessary comfort. In this study, university students participated as subjects, under the composite climatic of India, considering ASHRAE Class II protocols. The following is a summary of the findings from the analysis of the collected data:

1. The mean thermal sensations for the students in the dormitory were recorded as "slightly cool", "cold", "slightly cool", and "slightly warm" during the autumn, winter, spring, and summer seasons. The subject's mean thermal sensation was skewed towards "slightly cool" (mean TS = $-0.15$; sd = $\pm1.37$) for the combined dataset.
2. A total of 39.5%, 19.9%, 25.5%, and 40.5% of subjects in the hostel dormitories voted for "no change" in the persisting indoor thermal environment during the autumn, winter, spring, and summer seasons. However, 39.2%, 65%, and 31.2% of subjects preferred a warm thermal environment in the autumn, winter, and spring seasons, respectively. In comparison, about 13% of students preferred a cooler thermal environment in the combined dataset.
3. The PMV$-$PPD model overestimated and underestimated the actual thermal sensations in the summer and winter seasons.
4. The mean $T_{comf}$ was about $26.9 \pm 2.68$ °C, $19.9 \pm 2.11$ °C, $22.4 \pm 3.2$ °C, and $29.5 \pm 2.6$ °C for the autumn, winter, spring, and summer seasons, respectively.
5. Mean clo values of 0.57 ($\pm0.25$) clo, 0.98 ($\pm0.12$) clo, 0.45 ($\pm0.27$) clo, and 0.36 ($\pm0.11$) were recorded in the autumn, winter, spring and summer seasons, respectively. An average clothing value of about 0.49 (($\pm0.31$) clo was recorded for the pooled dataset, closely matching with the ASHRAE Standard 55 recommended clo value for the summer season.
6. More than 80% of subjects responded that they were comfortable when ceiling fans were operating. In contrast, only 60% of the subjects voted being comfortable when the windows were open at the survey time.

The study put forth the idea of future studies involving subjects of university dormitories regarding their comfort expectations and the use of environmental controls in different climates and geographical locations. An effective quantification of their thermal adaptation behavior and its impact on the comfort parameters will be advantageous for improving the students' thermal comfort and overall indoor thermal environment. It is also anticipated that the findings of this study will help building designers, architects, and engineers in designing energy-efficient and comfortable university hostel dormitories in the near future.

**Author Contributions:** Data curation, S.K., M.K.S., B.S.A., M.A.A. and N.A.-T.; Formal analysis, S.K., M.K.S. and B.S.A.; Methodology, S.K., M.K.S. and M.A.A.; Supervision, M.K.S.; Visualization, S.K., M.K.S. and N.A.-T.; Writing—original draft, S.K.; Writing—review & editing, M.K.S., B.S.A., M.A.A. and N.A.-T. All authors have read and agreed to the published version of the manuscript.

**Funding:** This research did receive any funding from any sources.

**Institutional Review Board Statement:** This is certified that the field surveying to study the thermal comfort expectation of subjects residing in hostel dormitories, located at Dr. B R Ambedkar National Institute of Technology, Jalandhar (Punjab) 144011, India, was carried out during the academic year 2018–2019. The field monitoring and subjective study doesn't violate any ethics and was fully in accordance with the data collection procedure laid down by the institute as well as International standard i.e. ASHRAE Standard 55-2013.

**Informed Consent Statement:** A brief introduction and discussion were provided to subjects by surveyor regarding the objective of the present study. A brief discussion with the occupant's also included the information about the type of data to be collected. However, no data has been collected regarding their personal information. The consent was also taken from each subject for the use of collected data for academic and academic research only. Same statement is also made by the authors in the acknowledgement section of the manuscript.

**Data Availability Statement:** The data collected in the study is available with the authors. Since authors are working on another project and present data will be a part of grand data base, authors would not like to make the data public at present. But if any researcher finds thae study interesting and requires some specific information and data for academic research then he/she can contact the corresponding author at the given e-mail address and the authors will be happy to help the researcher by providing the data and requested information.

**Acknowledgments:** Despite their class schedule, the author thanks all the subjects for participating in the field surveys. The authors also acknowledge the support received from authorities of Dr. B R Ambedkar National Institute of Technology, Jalandhar, towards the successful completion of the study. The authors would like to mention that this study did not receive any funding from any sources.

**Conflicts of Interest:** Corresponding author has informed all co-authors about this manuscript and took their permission before uploading it to the Sustainability journal. Corresponding author also like to state that there is no conflict of interest.

**Appendix A**

**Figure A1.** Questionnaire used in the study.

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
