# Peer review of "Investigation on Subjects’ Seasonal Perception and Adaptive Actions in Naturally Ventilated Hostel Dormitories in the Composite Climate Zone of India"

_sustainability, doi:10.3390/su14094997_

Round 1

Reviewer 1 Report

Investigation on subject’s seasonal perception and adaptive actions in naturally ventilated hostel dormitories in the composite climate zone of India

Page 1: “The large chunk of the energy consumed in the buildings to restore thermal comfort.” A verb is missing.

Good observation:

“Over the last two and half decades, numerous field studies carried out by researchers showed that PMV/PPD approach fails to capture the entire spectrum of parameters associated with psychological, psysiological (maybe physiological) and socio-cultural aspects.”

This is true, PMV/PPD approach fails. However, in the paper, your approach is about psychological, physiological and socio-cultural aspects, and nothing new is found.

Literature review is realized in the same manner. Nothing interesting is present.

Very trivial, non-useful words: “Generally, hostel dormitories are considered similar to residential buildings. However, the students in dormitories buildings carry out activities mostly related to relaxing, sitting, sleeping and doing paperwork-related learning and skill development. Therefore, the activities performed by students in dormitories are quite different from activities carried out in residential buildings offices and offices [18, 21].”

It is admirable the effort to follow for a year a large number of subjects to be interviewed. Of course, statistical data can be obtained, with interesting graphs, but what are the results?

The main result will be the “adaptive actions” (6):

“It can be seen that when students used the available controls such as opening windows, doors and switching on ceiling fans, the thermal comfort of occupants during different seasons was improved significantly. Also, during the summer season, it was observed that more than 80% of students voted comfortable when ceiling fans were operating whereas only 60% of the students voted comfortable when windows were open at the time survey.”

This is why this paper must be rejected.

Author Response

Dear Reviewer,

We appreciate the time and efforts put in by the reviewer in reviewing this manuscript. Thanks to the reviewer for his valuable comments and suggestions. We found them helpful in improving the quality and readability of the article. Based on the comments and suggestions, we have carefully made modifications to the revised manuscript. Please see our point-to-point responses to all your comments below, the original comments are in blue colour, and our responses are in black fonts. The corresponding corrections are incorporated and can be seen in the track change mode of the revised manuscript.

Thank you again for your time and consideration.

Yours sincerely,

Dr. Manoj Kumar Singh

Reviewer 2 Report

(1) How were the error bars calculated, starting from the bar charts in Fig. 3? What does 95% Cl mean?

(2) It is helpful to the readers to know the structure of the questionnaire. For example, in Fig. 4, the students in the survey can choose only 1 option, I guess. They are either comfortable in winter, or uncomfortable in an another season.

Author Response

(The authors gave the same response as above.)

Reviewer 3 Report

The following suggestion and comments should be taken

  • The abstract is well written as it contains a general overview of the problem statement, a summary of the methodology, and significant findings of the current research. However, the manuscript is hampered by several syntax and grammatical errors. Please kindly revise.
  • The objective could be combined. Not necessary to be two different objectives since reflected as the novelty of the study. Otherwise, why India has been chosen as a study area?
  • Figure 1. At the end of summer, it could be more interesting to be discussed since the intersect point of relative temperature (26.4) and relative humidity (64) obtain different peaks compared to other Months/Seasons.
  • Figure 4. What is the difference between comforts and un-comfort votes? Kindly clarify in detail.
  • Section 4.2. Please define more on the adaptive behavior. 
  • Figure 9. Did the author consider the daily weather? Such as rainy, sunny, and cloudy?
  • “..indoor air temperature was about 34°C…” How the author did fix the indoor air temperature? It is constant for the whole day?

Author Response

(The authors gave the same response as above.)

Round 2

Reviewer 1 Report

The authors responded to the observations to a large extent. I noticed some progress, especially in the argument from the literature review. 

Author Response

Thanks to the reviewers for their valuable comments and suggestions. We found them helpful in improving the quality and readability of the article. Based on the comments and suggestions, we have carefully made modifications to the revised manuscript. The corresponding corrections are incorporated and can be seen in the track change mode of the revised manuscript. 

Reviewer 2 Report

(1) I guess the author should revise their manuscript, at least provide more information to a general reader. The CI stands for confidence interval, which should be introduced either at its first appearance or use the full term. I have taken the probability or statistics classes before, so I can guess the possible meaning. For a general reader, it may be difficult to understand if math or physics is not the major.

Another issue is that why the 95% matters in evaluating the errors. You have a field study by selecting a certain number of people in a certain area to answer the questionnair and processing all the data. I don't think this research design is appropriate. You can group the people under survey and then calculate the averages and standard deviations to get the error bars. I do not think all the bar charts can have the same 5% erroneous. Therefore, I cannot agree with the authors that the investigation methods in Fig. 3 and Fig. 4 are believable and reliable. 

(3) Most importantly, the authors should also underline the impact of their study. Why can such questionair-based study provide any energy efficient power scheme? How do the results impact the energy efficiency and sustainability? What solutions can we derive from the survey results?

Author Response

(The authors gave the same response as above.)

Reviewer 3 Report

The authors made some revisions to the manuscript. However, some issues should be addressed. I recommended that the manuscript be accepted for publication after minor changes. 

Specific comments are provided as follows:

  • Please standardize font type.
  • Kindly revised the format references. Follow the guideline given in the website.

Author Response

(The authors gave the same response as above.)

Round 3

Reviewer 2 Report

The authors have made significant revision to the new version.